# Transcriptomic Analysis Reveals the Detoxification Mechanism of *Chilo suppressalis* in Response to the Novel Pesticide Cyproflanilide

**DOI:** 10.3390/ijms24065461

**Published:** 2023-03-13

**Authors:** Jungang Zhou, Lin Qiu, Qiyao Liang, Yi Zhou, Jinjun Sun, Qiao Gao, Hualiang He, Wenbing Ding, Jin Xue, Youzhi Li

**Affiliations:** 1College of Plant Protection, Hunan Agricultural University, Changsha 410128, China; 2Hunan Provincial Key Laboratory for Biology and Control of Plant Diseases and Insect Pests, Hunan Agricultural University, Changsha 410128, China; 3Hunan Provincial Engineering & Technology Research Center for Biopesticide and Formulation Processing, Hunan Agricultural University, Changsha 410128, China; 4National Research Center of Engineering & Technology for Utilization of Botanical Functional Ingredients, Hunan Agricultural University, Changsha 410128, China

**Keywords:** *Chilo suppressalis*, cyproflanilide, P450, RNA interference

## Abstract

*Chilo suppressalis* is one of the most damaging rice pests in China’s rice-growing regions. Chemical pesticides are the primary method for pest control; the excessive use of insecticides has resulted in pesticide resistance. *C. suppressalis* is highly susceptible to cyproflanilide, a novel pesticide with high efficacy. However, the acute toxicity and detoxification mechanisms remain unclear. We carried out a bioassay experiment with *C. suppressalis* larvae and found that the LD_10_, LD_30_ and LD_50_ of cyproflanilide for 3rd instar larvae was 1.7 ng/per larvae, 6.62 ng/per larvae and 16.92 ng/per larvae, respectively. Moreover, our field trial results showed that cyproflanilide had a 91.24% control efficiency against *C. suppressalis*. We investigated the effect of cyproflanilide (LD_30_) treatment on the transcriptome profiles of *C. suppressalis* larvae and found that 483 genes were up-regulated and 305 genes were down-regulated in response to cyproflanilide exposure, with significantly higher *CYP4G90* and *CYP4AU10* expression in the treatment group. The RNA interference knockdown of *CYP4G90* and *CYP4AU10* increased mortality by 20% and 18%, respectively, compared to the control. Our results indicate that cyproflanilide has effective insecticidal toxicological activity, and that the *CYP4G90* and *CYP4AU10* genes are involved in detoxification metabolism. These findings provide an insight into the toxicological basis of cyproflanilide and the means to develop efficient resistance management tools for *C. suppressalis*.

## 1. Introduction

*Chilo suppressalis* (Walker) (Lepidoptera: Crambidae) is one of China’s most damaging rice pests, with climate change, cropping systems and cultivar changes all contributing to increasingly frequent outbreaks [1,2,3]. Additionally, extensive pesticide use and improper application practices have been significant contributors to *C. suppressalis* resistance, resulting in severe economic losses due to outbreaks in rice-growing areas [4,5]. There is growing evidence that *C. suppressalis* has developed resistance to most pesticides, including nereistoxin insecticides (monosultap), organophosphorus insecticides (triazophos, chlorpyrifos), microbial pesticides (avermectin) and bisamide (cyantraniliprole, flubendiamide and chlorantraniliprole) [6,7]. Furthermore, recent studies have revealed that *C. suppressalis* has developed moderate resistance to cyantraniliprole (RR:77.6 folds), flubendiamide (RR:42.6 folds), triazophos (RR:64.5-461.3 folds) and chlorpyrifos (RR:10.1-125 folds) [4,7,8]. Hence, the development of new pesticides is crucial in the fight against *C. suppressalis*.

Cyproflanilide is a novel meta-diamide insecticide with toxicological activity against lepidopteran, coleoptera and thysanoptera pest species. It was classified into Group 30 “GABA-Gated Chloride Channel Allosteric Modulators” by the Insecticide Resistance Action Committee (IRAC). Several studies have identified metabolic and target resistance in *C. suppressalis* as resistance mechanisms to diamide insecticides [1,7,9,10,11,12]. Huang et al. (2020) reported that resistance to diamide insecticides was associated with a mutation of the ryanodine receptor (RyRs) at the I4578M, Y4667D/C, G4915E and Y4891F sites in *C. suppressalis* [8]. Furthermore, the overexpression of CYP genes (*CYP6CV5*, *CYP9A68*, *CYP321F3*, *CYP324A12*, *UGT40AL1* and *UGT33AG3*) were found to be involved in the metabolic resistance of *C. suppressalis* to chlorantraniliprole. [1,10] Cyproflanilide has been identified as playing an important role in pest resistance management because it binds to different insect receptors and induces toxicity via different modes of action than other diamide insecticides. However, the underlying mechanisms of *C. suppressalis* resistance remain unknown.

The ability of insects to metabolize and detoxify pesticides is thought to play a significant role in pesticide resistance. The detoxification enzyme genes involved in xenobiotic metabolism, degradation and detoxification belong to three distinct pathways: phase I (hydrolysis and oxidation-reduction), phase II (conjugation) and phase III (transport) [13,14]. Several important enzymes are involved in the metabolism of xenobiotics, including cytochrome P450 monooxygenase (P450s), esterase (EST, including carboxylesterase and phosphoesterase), glutathione S-transferase (GST), UDP-glycosyltransferase (UGT) and ATP-binding cassette transporter (ABC) [15,16]. P450 belongs to the heme family of proteins involved in the metabolism of endogenous and exogenous compounds and the regulation of insect growth and development [15,17]. It is worth noting that the CYP gene plays a central role in insect xenobiotic metabolism, hydrolysis and oxidation-reduction and regulates a variety of endogenous signaling molecules [16]. However, we do not know whether CYP genes are involved in the metabolism of cyproflanilide.

To evaluate the control efficiency of cyproflanilide on *C. suppressalis*, we carried out a laboratory bioassay and field trial. Our results revealed that cyproflanilide has effective toxicological insecticidal activity against *C. suppressalis*. Furthermore, in order to find the detoxification metabolism genes that were mainly involved in cyproflanilide, a combined transcriptome RNA-seq and RNA interference approach identified two P450 genes, *CYP4G90* and *CYP4AU10*, that are involved in cyproflanilide metabolism in *C. suppressalis*. Our findings contribute to a better understanding of cyproflanilide metabolism in *C. suppressalis* and provide insight into the development of pest and resistance management strategies for this species.

## 2. Results

### 2.1. Bioassay Results

Bioassay experiments showed that cyproflanilide exhibited potent pesticide activity against *C. suppressalis*. The probit analysis showed that LD_10_, LD_30_ and LD_50_ were 1.421, 5.261 and 13.022 ng/per larva at 24 h (Table 1). The third instar larvae were then treated with LD_10_, LD_30_ and LD_50_ cyproflanilide to test the expression of detoxification genes.

### 2.2. Field Experiment Results

After 21 days of spraying, compared to 200 g/L chlorantraniliprole (SC), the most effective pesticide for controlling *C. suppressalis* was 20% cyproflanilide (SC). The control efficiency on *C. suppressalis* of 16.2, 32.4, 48.6 and 64.8 g/hectare of 20% cyproflanilide (SC) were 77.38%, 84.60%, 87.26% and 91.24%, respectively (Table 2). There was a significant difference in control efficiency among low, medium and high doses (*F* = 74.941, *p* = 0.0001). However, there was no significant difference in control efficiency between 48.6 and 64.8 g/hectare of 20% cyproflanilide (SC) (Table 2). The dead heart rate was significantly reduced when rice was treated with different doses of 20% cyproflanilide (SC) for 21 days compared to 200 g/L chlorantraniliprole (SC).

### 2.3. Transcriptome Analysis

#### 2.3.1. Sample Quality Control and Correlation Analysis of 6 Samples

The six individual cDNA libraries constructed from the contrast and cyproflanilide treatments were sequenced on Illumina Nova Seq 6000. Clean reads for subsequent analysis were obtained after filtration of the original data, a sequence error rate check and a GC content distribution check. A total of 36.58 GB of clean data was obtained from the RNA sequences of six samples. The GC content of the sequence data ranged from 46.29% to 46.83%, and the Q20 and Q30 ratios were > 97% and 93%, respectively, indicating good quality (Appendix A).

Pearson’s correlation coefficient was used to analyze the relationship between the control and treatment groups. The square of coefficient was > 0.8 (Appendix A), indicating that the within-group biological replication was good. Gene expression levels (FPKM values) of each sample were used to calculate the within and between group correlation coefficients which were then plotted into a heat map (Figure 1a).

#### 2.3.2. GO and KEGG Analysis of DEGs

GO pathway enrichment analysis was performed to investigate the function of DEGs in *C. suppressalis* third instar larvae exposed to cyproflanilide. Regarding the 19 significant GO terms, 393 DEGs were enriched, of which 339 were up-regulated and 54 were down-regulated. The 19 significantly enriched GO terms were divided into three categories: molecular functions, biological processes and cellular components. The most abundant genes in the molecular function terms category were “structural molecular activity” and “structural constituent of cuticle”. For the biological processes category, the most abundant of genes were “carbohydrate derivative metabolic process” and “drug metabolic process”. The “extracellular region” had the most genes in the cellular component category. Down-regulated genes were enriched in DNA replication. Additionally, “drug metabolic process”, “serine hydrolase activity” and “oxidoreductase activity” may also be related to cyproflanilide exposure (Figure 1b and Appendix A).

The results of the enrichment analysis revealed that “DNA replication” was significantly enriched in 101 KEGG pathways (Appendix A). The abundant DEGs found in the KEGG database mainly included categories such as “pentose and glucuronate interconversions”, “ubiquinone and other terpenoid-quinone biosynthesis”, “drug metabolism-other enzymes”, “fatty acid biosynthesis” and “base excision repair”. It is worth noting that “drug metabolism-other enzymes”, “drug metabolism-cytochrome P450” and “metabolism of xenobiotics by cytochrome P450” may also be involved in the metabolism of cyproflanilide. We found that the detoxification metabolism pathway was significantly enriched, indicating that these pathway genes were involved in detoxification metabolism to cyproflanilide.

#### 2.3.3. Transcriptome Profiling Reveals Detoxification Genes Associated with Cyproflanilide

Using a log2 fold change cutoff of ≥ 1 and ≤ − 1, and corrected *p* value of < 0.05, a total of 788 genes were identified as differentially expressed between the two treatments, of which 483 genes were up-regulated and 305 down-regulated (Appendix A). Finally, the analysis of DEGs identified 28 detoxification metabolism genes, including 3 glutathione S-transferase, 4 glucosyltransferase, 11 cytochrome oxidoreductase, 5 carboxylesterase and 5 ATP-binding cassette transporters. qRT-PCR was performed on the 12 metabolic detoxification genes to check if there were any differences in the transcriptome data. The qRT-PCR results show a correlation between the metabolic detoxification gene expression level and transcriptome data.

### 2.4. Analysis Detoxification Gene Expression Level by qRT-PCR

We used qRT-PCR to see if the 12 detoxifications genes differed with different pesticide concentrations, and over time. At 24 h, *UGT33AG3*, *EST5* and *CYP6AB49* were significantly different under the stress of three sublethal doses (Figure 2a,c and Figure 3a). *ABCC3* and *EST17* had significantly different levels under LD_10_ stress, while *EST17* was also significantly different under LD_30_ stress (Figure 2b,c). *ABCC6* was significantly different under LD_10_ and LD_50_ stress (Figure 2b). *CYP6AB48* and *CYP367A9* were significantly different under LD_50_ and LD_30_ stress (Figure 3a). At 48 h, *ABCC3*, *CYP6AB49* and *CYP6AB48* had significantly different levels only under LD_50_ stress (Figure 2e and Figure 3b). *CYP321F1* was significantly different under the stress of three sublethal doses (Figure 3b). The expressions of *EST17* and *CYP367A9* were inhibited under the stress of three sublethal doses (Figure 2f and Figure 3b). *UGT33AG1* was significantly different only under LD_30_ stress. *UGT33AG3*, *ABCC6* and *EST5* had significantly different levels under LD_10_ stress, while *ABCC6* was also significantly different under LD_30_ stress (Figure 2d–f). Interestingly, the results of the gene expression level showed that *CYP4G90* and *CYP4AU10* have stable up-regulation expression levels at different concentrations and across time (Figure 3a,b).

### 2.5. Sensitivity to Pesticides after Gene Silencing

The high expressions levels of *CYP4G90* and *CYP4AU10* at different cyproflanilide concentrations and exposure times suggest their likely involvement in cyproflanilide metabolism. Hence, RNAi was used to verify the functions of *CYP4G90* and *CYP4AU10* in *C. suppressalis*. After 24 h and 48 h of feeding on a diet containing dsRNA specific for *CYP4G90* and *CYP4AU10*, the mRNA level decreased by 65%, 46% and 44%, 42% (Figure 4a,b), respectively. When larvae were treated with LD_30_ cyproflanilide for 12 h, the knockdown of *CYP4G90* and *CYP4AU10* significantly increased the mortality rates of third instar larvae compared to the control groups (fed dsEGFP, Figure 4c). After 24 h, the knockdown of *CYP4G90* during LD_30_ cyproflanilide treatment resulted in a significant increase in the mortality of third instar larvae compared to the control groups (Figure 4d). In contrast, the knockdown of *CYP4AU10* had no significant effect on the larvae mortality of LD_30_ cyproflanilide treatment in 24 h (Figure 4d). Collectively, these results demonstrate that a decrease in *CYP4G90* and *CYP4AU10* mRNA levels can result in an increase in the mortality of larvae exposed to cyproflanilide, further confirming the involvement of *CYP4G90* and *CYP4AU10* in the metabolism of cyproflanilide.

## 3. Discussion

Due to their high efficiency, chemical pesticides such as monosultap, triazophos, avermectin and chlorantraniliprole are currently the most effective method of controlling large-scale outbreaks of *C. suppressalis* [6,7]. Chlorantraniliprole sales in China have increased significantly due to the increasing number of *C. suppressalis* outbreaks, resulting in a significant increase in resistance level [18,19]. The new insecticide cyproflanilide is highly effective against *C. suppressalis* larvae. Over time, lethal doses of the insecticide in the field will gradually reduce to sublethal doses, which will continuously threaten larval growth and development.

Sublethal doses of insecticides have been reported to affect development duration, pupal weight and reproduction of pests. For example, chlorpyrifos, etofenprox and phosmet prolonged larval development time, and chlorantraniliprole and flubendiamide increased larval and pre-pupal development times and decreased larval weight [20,21]. During physiological activity, carbohydrate and lipid metabolism, and energy metabolism are the primary source of energy and substrate for organisms [22]. In this study, 7 out of 12 DEGs involved in carbohydrate metabolism were down-regulated, indicating that it may influence *C. suppressalis* development. Interestingly, under chlorantranilamide treatment, these pathway genes were also altered which affected the development time of *C. suppressalis* larvae [22,23]. Lipids play an important role in insect reproduction [24,25,26], and studies have demonstrated that changes in lipid content affect silkworm reproduction [27]. In our study, 31 DEGs were associated with glycerolipid metabolism, sphingolipid metabolism, steroid biosynthesis, fatty acid elongation, glycerophosholipid metabolism, fatty acid degradation, fatty acid biosynthesis, fatty acid metabolism and unsaturated fatty acid biosynthesis.

In addition to the effect of exogenous substances on insect reproduction and development, the detoxification and metabolism of exogenous substances by pests have been extensively studied [28,29,30,31]. In this study, 28 DEGs were primarily enriched in the “drug metabolism- other enzymes”, “glutathione metabolism” and “Metabolism of xenobiotics by cytochrome P450” pathways. P450 genes, in particular, have received increasing attention due to their broad range of functions [32]. For example, the overexpression of *CYP4G19* in beta-cypermethrin-resistant strains of *Blattella germanica* was positively correlated with a high level of cuticular hydrocarbons, while the knockdown of *CYP4G19* expression resulted in a decrease in cuticular hydrocarbons and reduced insecticide tolerance in resistant strains [33]. Furthermore, previous studies have reported a link between the function of the CYP4 family of genes and the insect cuticle [17,33,34,35]. In this study, we demonstrated that *CYP4G90* and *CYP4AU10* are involved in cyproflanilide metabolism, and we found a number of DEGs related to the insect cuticle. Therefore, *C. suppressalis* may develop cuticular resistance to cyproflanilide. Transcriptome analysis of *C. suppressalis* treated with chlorantraniliprole revealed that *CYP4G90* is also a DEG. However, there was no evidence to suggest that *CYP4G90* was involved in chlorantraniliprole metabolism [22]. A significant increase in *UGT33AG3* expression was found in *C. suppressalis* strains resistant to chlorantraniliprole [10]. Our results showed a significant difference in *UGT33AG3* expression levels following 24 h of cyproflanilide treatment. It is evident that the development of pest resistance is not dependent on a single gene [36,37,38]. Previous research using RNAi demonstrated that *CYP6CV5*, *CYP9A68*, *CYP321F3* and *CYP324A12* are involved in metabolic resistance to chlorantraniliprole [1].

In this study, *CYP4G90* and *CYP4AU10* had higher expression levels compared with other detoxifying metabolic genes under the cyproflanilide treatment of *C. suppressalis* third instar larvae. Meanwhile, we found that knockdown of these two genes increased the mortality of *C. suppressalis* larvae exposed to cyproflanilide. In the reports of the resistant populations of insects to pesticides, the overexpression of P450 genes were related to the metabolic resistance of pesticides. For example, *CYP321A6* and *CYP332A1* had mediate chlopyrifos resistance in *Spodoptera exigua*, *CYP6BG1* was involved in the resistance of imidacloprid to whiteflies [39,40] and *CYP6CM1* and *CYP4C62* were proven to have participated resistance to chlorantraniliproe and chlorpyrifos in *Plutella xylostella* and *Nilaparvata lugens*, respectively, by RNAi [41,42]. In future, with the wider use of cyproflanilide, the *C.suppressalis* may develop resistance to it. Therefore, these two genes are the potential target resistance genes of *C. suppressalis* to cyproflanilide.

In conclusion, the novel insecticide cyproflanilide is highly effective against *C. suppressalis*, with a control efficiency of 64.8 g/hectare reaching 91.24% efficacy. We also demonstrated the role of *CYP4G90* and *CYP4AU10* in the metabolic detoxification of cyproflanilide. Our findings show that the novel pesticide cyproflanilide is highly effective against *C. suppressalis* and we identified potential targets for further pesticide resistance research into cyproflanilide.

## 4. Materials and Methods

### 4.1. Insect and Insecticide

The *C. suppressalis* strain used in this study was collected from a field population in Yongan, Hunan province, China, during 2021. All larvae were maintained at 28 ± 1 °C under 60–70% relative humidity (RH) and a 16 h: 8 h (light/dark) photoperiod. Larvae were fed on *Zizania latifolia* (Griseb). Cyproflanilide (98.91% active ingredient) and cyproflanilide SC (20% active ingredient) were provided by Nantong Taihe Chemical Co., Ltd. (Nantong, Jiangsu, China). Chlorantraniliprole was purchased from Fumeishi China Investment limited (Shanghai, China).

### 4.2. Bioassay

The third instar larvae were used for the bioassay in this research. Each treatment was repeated thrice, with 30 larvae per replicate. Cyproflanilide was dissolved in acetone to obtain concentrations of 40 mg/L, 20 mg/L, 10 mg/L, 5 mg/L, 2.5 mg/L, 1.25 mg/L and 0.625 mg/L, with the acetone-treated group as a control. *C. suppressalis* larvae were treated with 0.5 µL of the insecticide solution drop on the pronotum using a hand micro applicator (Burkard Manufacturing Co Ltd., Rickmansworth, Hertfordshire, England), and mortality was recorded at 24 h. The sublethal dose of cyproflanilide to the *C. suppressalis* was determined via a log-probit analysis of bioassay data using SPSS 22. Larvae treated with LD_10_, LD_30_ and LD_50_ were collected at 24 h and 48 h, and were immediately frozen in liquid nitrogen and then stored at −80 °C for RNA extraction.

### 4.3. Evaluting and Verifying Field Efficacy

The field-plot trials were located in Xidu Town (Hengyang County, Hunan Province, China). When the treatment rice plant was applied, most of the pests were second and third instar larvae. The field experiments were conducted to evaluate the control efficacies of cyproflanilide and chlorantraniliprole against *C. suppresalis* larvae on rice plants. All of the experiments were designed as randomized complete blocks with four replicates of each treatment. The 7.5 × 4 m plots were used for each treatment application. The following four treatments of 20% cyproflanilide (SC) and one treatment of 200 g/L chlorantraniliprole (SC) were evaluated: 16.2 g/hectare, 32.4 g/hectare, 48.6 g/hectare 64.8 g/hectare and 30 g/hectare, with water as a control. After 21 days of pesticide application, we counted the number of rice dead hearts caused by *C. suppressalis*. The number of rice dead hearts was used to calculate the insecticide control effect. No other pesticides were applied during the experimental period. The average total number of rice plants in the plot was obtained according to the number of rows and columns of rice in the plot and the average tiller number of 50 clusters. The control efficacies of each insecticide-treated group *C. suppressalis* larvae were calculated using the following Equations (1) and (2):Dead heart rate (%) = (The number of dead rice hearts / Total number of rice plants in the experimental plot) × 100.(1)
Control efficiency (%) = (Number of dead rice hearts of control plants − Dead heart rate of pesticide-treated plants/ Dead heart rate of control) × 100.(2)

### 4.4. RNA Sequencing and Annotation of Unigenes

Total RNA was used as the input material for the RNA sample preparations. Briefly, mRNA was purified from total RNA using poly-T oligo-attached beads. Fragmentation was carried out using divalent cations under an elevated temperature in the First Stand Synthesis Reacting Buffer (5X). The first strand cDNA was synthesized using random hexamer primer and M-MuLV Reverse Transcriptase; we then used RNaseH to degrade the RNA. Second strand cDNA synthesis was subsequently performed using DNA Polymerase I and dNTP. The remaining overhangs were converted into blunt ends via exonuclease/polymerase activities. After adenylation of the 3′ ends of DNA fragments, Adaptor with a hairpin loop structure was ligated to prepare for hybridization. To select cDNA fragments of preferentially 370~420 bp in length, the library fragments were purified with the AMPure XP system (Beckman Coulter, Beverly, USA). After PCR amplification, the PCR product was purified using AMPure XP beads, and the final library was obtained.

The library was initially quantified using a Qubit2.0. Fluorometer, then diluted to 1.5 ng/µL. The insert size of the library was detected using an Agilent 2100 bioanalyzer. After the insert size met expectations, qRT-PCR was used to accurately quantify the effective concentration of the library (the effective concentration of the library is higher than that of 2 nM) to confirm the quality of the library. The different libraries were then pooled according to the effective concentration and the target amount of data of the machine before being sequenced using the Illumina NovaSeq 6000. The reference genome and gene model annotation files were downloaded from a genome website directly: http://v2.Insect-genome.com/api/Download/.-01_data-01_speciesChilo-suppressalis-Chilo_suppressalis.genome.fa (accessed on 11 August 2021). The index of the reference genome was built using Hisat2 (v2.0.5) and paired-end clean reads were aligned to the reference genome using Hisat2 (v2.0.5). Transcripts with an adjusted *p*-value < 0.05 found by DESeq were assigned as differentially expressed.

### 4.5. Quantitative Real Time PCR

First-stand cDNA templates were synthesized using the PrimeScriptTM RT reagent kit with gDNA Eraser (Taraka, Dalian, China). The specific primers for qRT-PCR were designed using online website NCBI: https://www.ncbi.nlm.nih.gov/tools/primer-blast/ (accessed on 27 December 2021) with the EF-1 house-keeping gene used as the internal gene (Appendix A) [43]. The qRT-PCR reaction was performed using Hieff^®^ qPCR SYBR Green Master Mix (Yeasen, Shanghai, China) following the manufacturer’s instructions. The 10 µL PCR reaction volume contained 5 µL SYBR Green Master Mix, 2.2 µL diethylpyrocarbonate-treated water (DEPC), 2 µL diluted cDNA template with a concentration of 100 ng/µL, and 0.4 µL of each primer. The qRT-PCR program was as follows: 95 °C for 30 s, 40 cycles of 95 °C for 5 s and 60 °C for 30 s. The relative gene expression levels were represented using the 2^−ΔΔCT^ method.

### 4.6. RNAi Experiment

Primer for RNA interference (RNAi) was designed using online website siDirect: http://sidirect2.rnai.jp/ (accessed on 27 March 2022) and Vazyme: https://crm.vazyme.com/cetool/en-us/singlefra-gment.html (accessed on 27 March 2022). The templates of double-stranded RNA (dsRNA) synthesis were obtained by the real-time polymerase chain reaction (RT-PCR) using specific primers (Appendix A). These two genes, a 528 bp fragment of *CYP4G90* and a 553 bp fragment of *CYP4AU10*, were amplified and subcloned into the pET-2p expression vector using the ClonExpress II One Step Cloning Kit (Vazyme, Nanjing, China). The recombinant vectors of *CYP4G90* and *CYP4AU10* were transformed into HT115-competent cells (Shanghai Weidi Biotechnology Co., Ltd., Shanghai, China) for dsRNA expression. Individual colonies were inoculated and grown until the cultures reached an OD_600_ of 0.8. Isopropyl-β-D-Thiogalactoside (IPTG) (Beijing coolaber Technology Co., Ltd., Beijing, China) was added to the cultures to produce a final concentration of 0.1 mM; the culture was then incubated at 37 °C for approximately 4 h. The expression of dsRNA was verified by 1% agarose gel. The induced cultures were centrifuged at 8000× *g* for 5 min before being resuspended in one-tenth of the original culture volume of 0.05 M phosphate-buffered saline (PBS). The resuspended bacterial solution was then used for oral RNAi. The third instar larvae were starved for 2 h, while *Zizania latifolia* (Griseb) was soaked in the resuspended bacterial solution and then dried for 30 min. After feeding on dsRNA for 24 h, larvae were treated with of 0.5 µL of the LD_30_ cyproflanilide solution. Each treatment was replicated five times with 30 third instar larvae per replication and the mortality was recorded every 12 h until the end of the 24 h period.

### 4.7. Data Processing and Statistical Analysis

We used GraphPad Prism8 and IBM SPSS Statistics Version 22 software for data analyses. T tests (nonparametric tests) were used to compare means between treatments and their respective controls. The results are given as means ± standard error. *p* < 0.05 was considered a significant difference, and *p* < 0.01 was considered a highly significant difference.

## Figures and Tables

**Figure 1 ijms-24-05461-f001:**
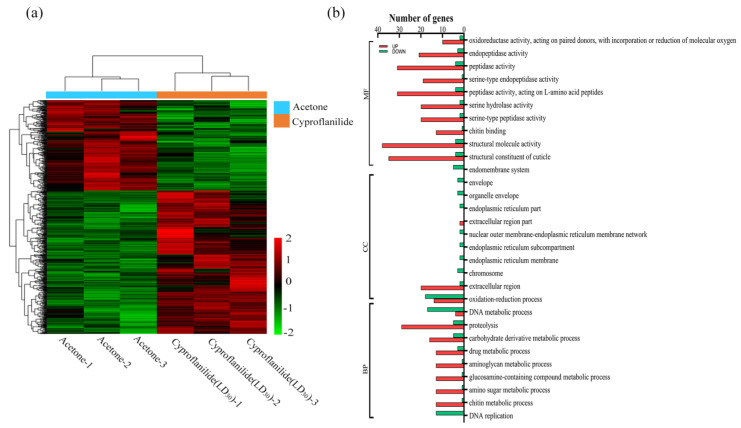
Heat map and GO classification of DEGs in the treatment (**a**): Cluster heat map of differentially expressed genes. The abscissa in the figure refers to the sample names and the ordinate is the normalized value of the differential gene FPKM. The higher the expression level, the redder the color, and the lower the expression level, the greener the color (**b**): The number of all differential genes in the transcriptome in the GO classification. Red indicates up-regulated genes and green indicates down-regulated genes. The x-axis displays the names of the pathways; the y-axis represents the number of genes enriched in the respective pathways.

**Figure 2 ijms-24-05461-f002:**
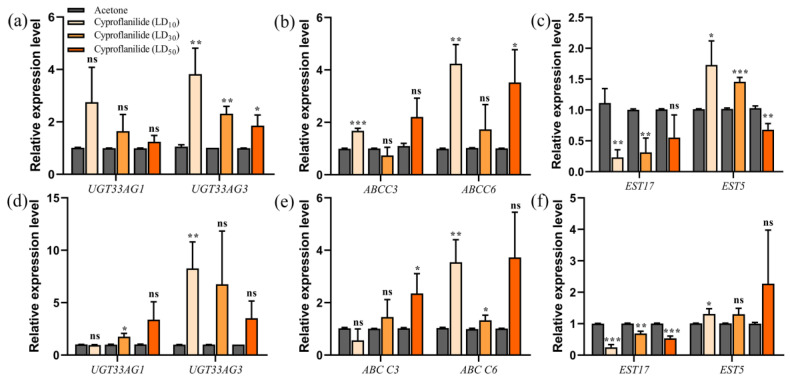
Expression levels of UGT, ABC and EST genes after cyproflanilide-treated larvae *C. suppressalis.* Relative expression levels following treatment durations of 24 h (**a**–**c**) and 48 h (**d**–**f**). Grey represents acetone treatment, the other colors represent different concentrations of cyproflanilide. * *p* < 0.05, ** *p* < 0.01, *** *p* < 0.001, ns = not significant.

**Figure 3 ijms-24-05461-f003:**
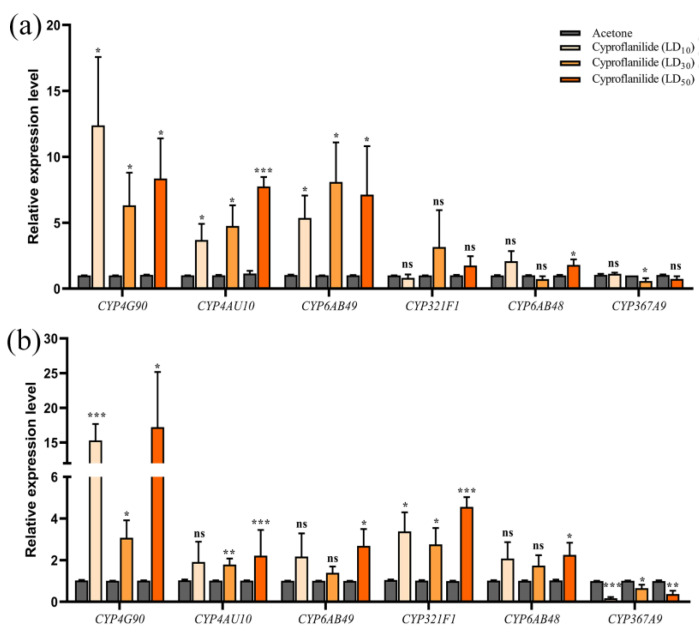
Expression level of CYP genes after cyproflanilide-treated larvae *C. suppressalis*. Relative expression level after 24 h (**a**) and 48 h (**b**) treatment duration. Grey represents acetone treatment, the other colors represent different concentrations of cyproflanilide. * *p* < 0.05, ** *p* < 0.01, *** *p* < 0.001, ns = not significant.

**Figure 4 ijms-24-05461-f004:**
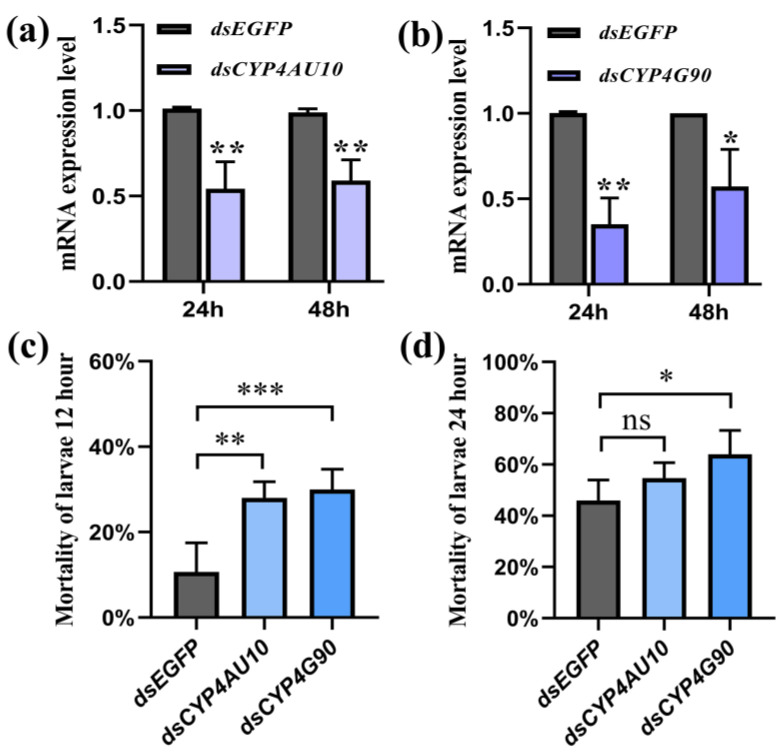
RNA interference efficiency and mortality. (**a**,**b**): RNA interference efficiency. The number of deaths at 12 h (**c**) and 24 h (**d**) post-LD_30_ treatment following RNAi. **p* < 0.05, ** *p* < 0.01, *** *p* < 0.001, ns = not significant.

**Table 1 ijms-24-05461-t001:** Toxicity of cyproflanilide against the third instar larvae of *C. suppressalis* at 24 h after treatment.

Insecticide	Degree of Freedom	χ^2^	LD_10_ Value (95% Confidence Interval)	LD_30_ Value (95% Confidence Interval)	LD_50_ value (95% Confidence Interval)
Cyproflanilide solution	5	5.42	1.7 ng/per larva (1.034–1.8306)	5.2612 ng/per larva (4.3203–6.4241)	13.02 ng/per larva (10.3713–17.1871)

**Table 2 ijms-24-05461-t002:** The control efficiency of cyproflanilide (SC) difference in active ingredients against the field strain *C. suppressalis*.

Treatment Groups	Active Ingredient (g/hectare)	Control Effect after 21 Days (%)	Significance of Difference (5%)
20% Cyproflanilide SC	16.2	77.38 ± 2.54	b
	32.4	84.6 ± 5.09	ab
	48.6	87.26 ± 3.58	a
	64.8	91.24 ± 2.72	a
200 g/L Chlorantraniliprole^®^	30.0	35.17 ± 1.36	c

**Note:** Control efficacy significant differefnces based on Duncan’s multiple comparison test at *p* < 0.05 level. All data are represented means ± standard error (S.E.)

## Data Availability

Not applicable.

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
