# Peer review of "Transcriptomic Analysis Reveals the Detoxification Mechanism of Chilo suppressalis in Response to the Novel Pesticide Cyproflanilide"

_ijms, 2023, doi:10.3390/ijms24065461_

Round 1

Reviewer 1 Report

The manuscript by Zhou et al. describes the efficacy of cyproflanilide against C. suppressalis, the effect of the insecticide on transcriptome profiles, and the involvement of CYP4G90 and CYP4AU10 in detoxification of the insecticide. In the introduction, the authors stated ‘to evaluate the control effect of cyproflanilide’, but they need to explicitly state the study's aims. The authors used ‘control efficiency’ throughout to mean insecticidal efficiency/efficacy. There is also ‘control’ to be compared with ‘treatment’. It is not a major problem, but clearer terminology should be used. Overall, the manuscript has merits and deserves to be published.

Minor corrections.

Table 1, make significant figures of CL10, LD30, and LD50 consistent. Is there LD90 value measured?

In section 2.2 and Table 2, 20% cyproflanilide (SC) was used, which was compared to 200 g/L chlorantraniliprole (SC). Is there any reason to use different units of concentration? Isn’t 200 g/L also 20% solution? Make the concentration unit consistent if possible and specify w/v or v/v, etc.

Figure 1, x-axis texts and texts in 1b are hardly readable. Increase font size of all texts used in the figure.

Figure 2 and 3, legend text and colour are hardly readable. Increase the legend key and font to a reasonably visible size.

L191, remove ‘s’ in ‘expressions’

Reviewer 2 Report

In this manuscript, the authors reported the efficacy of a novel insecticide, cyproflanilide, against the Chilo suppressalis, and found the expression of the CYP4G90 and CYP4AU10 gene of the C. suppressalis was up-regulated by qRT-PCR when treated with cyproflanilide. The RNAi experiment also showed that the mortality changed when the CYP4G90 or CYP4AU10 gene were knocked down. There are some points need to be proved or explained.

Comments:

1 Line 99-100 What is the mean of “low, medium, and high doses”? Please specify the corresponding concentration.

2 Line 96-97 Why you choose “chlorantraniliprole” as a control pesticide?

3 What’s the mean by “may involve in”? Are those pathways be significantly enriched or not?

4 Line 157 Why you finally focus on 28 detoxification metabolism genes?

Minor revisions:

1 Line 19-20 In consecutive sentences, the same inflection words appear frequently.

2 Line 57 modify “cytochrome p450 monooxygenase(P450s)”

3 Line 65-66 It’s recommended to modify “have three distinct pathway” to “belong to three distinct pathway”.

4 Line 71 p450 should beP450s

5 Line 94 c should be c2

6 Line 107-108 It’s Duncan’s in stand of Dunn’s

7 Line 108 “All data represent means ± S.E.” should be “All data are represented as means ± S.E.”

8 Line 124 Each separate figure in Figure 1 should be marked with lowercase letters

9 Line 132 What is the purpose of GO and KEGG analysis? The specific results of these analyses need to be further stated.

10 Line 154 “Using a log2 fold change cutoff of ≥ 1” should be “Using a log2 fold change cutoff of ≥ 1 and -1”.

11 The results of data statistics analysis in table 2 should be added in the text (such as F value and P value).

12 Please unify the citation style of Figures. (such as in line 166 168 167 “Figure 3 a” or “Figure 3a”?)

13 Line 194-202 Those sentences may confuse the readers. Please simplify those sentences to make it more readable.

14 Line 215-217 “Chlorantraniliprole sales in China have increased significantly” revised to “Chlorantraniliprole was overused in China”

15 The hypothesis in line 235-236 is groundless.

16 Line 251 It is not appropriate to use “interestingly” for the following result.

17 The discussion in line 262-271 has nothing to do with the results of this manuscript, it is recommended to delete this part. The discussion part must be rewritten.
